# A comparison of presentations and outcomes of salvage versus non-salvage abdominal free flap breast reconstructions—Results of a 15-year tertiary referral centre review

Christine Bojanic[1]*, Bruno Di Pace[2,3], Dina T. Ghorra[1,4], Laura J. Fopp[1], Nicholas G. Rabey[1], Charles M. Malata[1,3,5]

1 Plastic & Reconstructive Surgery Department, Addenbrooke's Hospital, Cambridge University Hospitals NHS Foundation Trust, Cambridge, United Kingdom, 2 Scuola Superiore Meridionale, University of Naples "Federico II", Naples, Italy, 3 School of Medicine, Anglia Ruskin University, Cambridge and Chelmsford, United Kingdom, 4 Department of Plastic Surgery, University of Alexandria, Alexandria, Egypt, 5 Cambridge Breast Unit, Addenbrooke's Hospital, Cambridge University Hospitals NHS Foundation Trust, Cambridge, United Kingdom

* cb2097@cam.ac.uk

## Abstract

### Introduction

Salvage breast reconstruction with autologous tissue is becoming more prevalent due to a resurgence in implant-based procedures. The latter has caused a commensurate rise in failed or treatment-resistant prosthetic cases requiring conversion to free tissue transfers. Salvage reconstruction is often considered more challenging, owing to patient presentation, prior treatments and intraoperative difficulties. The aim of the study was to test this hypothesis by comparing outcomes of salvage versus non-salvage autologous microsurgical breast reconstructions in a retrospective matched cohort study.

### Methods

The demographics, risk factors, operative details and outcomes of patients who underwent free flap salvage of implant-based reconstructions by a single operator (2005–2019) were retrospectively evaluated. For each salvage reconstruction, the consecutive non-salvage abdominal free flap reconstruction was selected for comparison. The clinical outcomes including intraoperative blood loss, operative time, flap survival and complication rates were compared.

### Results

Of 442 microsurgical patients, 35 (8.0%) had salvage reconstruction comprising 41 flap transfers (29 unilateral, 6 bilateral) and 42 flaps (28 unilateral, 7 bilateral) in nonsalvage reconstruction. Deep inferior epigastric perforator (DIEP) flaps comprised the commonest autologous tissue used in both groups at 74% and 71% respectively. Most patients (83%) underwent salvage reconstruction for severe capsular contractures. There was a significant

**Data Availability Statement:** All relevant data are within the paper and its Supporting information files.

**Funding:** The author(s) received no specific funding for this work.

**Competing interests:** The authors have declared that no competing interests exist.

difference in radiation exposure between groups (salvage reconstruction 89%, non-salvage reconstruction 26%; p<0.00001). All 83 flaps were successful with similar reoperation rates and intraoperative blood losses. Unilateral salvage reconstruction took on average two hours longer than non-salvage reconstruction (p = 0.008). Overall complication rates were similar (p>0.05).

## Conclusion

This 15-year study shows that despite salvage autologous free flap breast reconstruction requiring longer operation times, its intra and postoperative outcomes are generally comparable to non-salvage cases. Therefore, salvage breast reconstruction with free flaps provides a reliable option for failed or suboptimal implant-based reconstructions.

## Introduction

Breast cancer is the most common malignancy in women, with over 2 million new cases each year worldwide [1]. Around 40% of these women will require a mastectomy as part of their treatment [2]. Following mastectomy, 21% of women in the UK and 43% of women in the US choose to undergo breast reconstruction [3–5]. Implant-based breast reconstruction is the most commonly performed type of breast reconstruction across the globe [6–8]. However, although reconstruction with prostheses is considered to be "simple" and has distinct advantages in certain patients, its limited lifespan and frequent unsatisfactory long-term patient outcomes (especially in the context of adjuvant radiotherapy) have resulted in a significant proportion of patients requiring multiple revision surgeries [7–9]. The number of patient referrals for failed implant-based cases or those with recalcitrant complications has been steadily increasing with implant removal now being the third most commonly performed breast reconstruction procedure in the US after implant-based reconstruction and breast reduction [10–12].

In cases where implant reconstruction is unsuccessful, suboptimal or ridden with intractable complications, a conversion to autologous tissue (usually a free flap) can be performed; this is termed salvage (or as originally described tertiary), breast reconstruction [4, 8, 9, 13, 16]. Salvage breast reconstruction has been deemed challenging for reasons mainly related to patient presentation and intra-operative challenges [7, 9]. These include an older patient demographic with increased co-morbidities, chronic or persistent implant-related pain, and frustration with their suboptimal or symptomatic reconstructions [3, 9]. Patients referred for salvage breast reconstruction also present with more complex issues such as irradiated scarred skin and soft tissues, severe capsular contracture (CC), silicone lymphadenopathy and scarred internal mammary recipient vessels [2, 9, 14, 15]. The intraoperative challenges include bleeding from partial or total capsulectomies, scarring interfering with the anastomoses, skin shortage and the need to manage both soft and hard capsules [16, 17]. Further problems include management of the pectoralis major muscle, repair of the often-disrupted inframammary fold as well as the correction of poor shape of the breast mound in the context of stiff distorted tissues [18]. More recently surgeons also have had to contend with extra scarring caused by previous acellular dermal matrix (ADM) incorporation into soft tissues where this was used as an adjunct to implant reconstruction [11, 18]. This can be in the context of ruptured or failed implants with silicone "extravasation" outside the capsules which create additional challenges [19]. It is interesting that extracapsular rupture has been estimated to be about 25.8% and

therefore this is a common practical problem [20]. In these complex salvage breast reconstruction cases, autologous free flap conversion has been the favoured method [2, 7, 15].

The objective of this study was to compare the presentation and outcomes of patients undergoing autologous free flap salvage of previous implant-based breast reconstruction to those of non-salvage free flap breast reconstruction performed over a cumulative period of 15 years by a single operator. To our knowledge, this is the first 15-year retrospective study that compares each salvage free flap breast reconstruction (SR hereafter) with a chronologically consecutive non-salvage autologous free flap breast reconstruction (NSR hereafter).

## Materials and methods

A retrospective review of all free flap breast reconstruction patients who underwent salvage of their previous prosthetic reconstructions by a single operator (CMM) at Cambridge University Hospital (Addenbrookes) between January 2005 and September 2019 was undertaken.

Patients were identified from a prospective departmental free flap register cross-referenced with the senior author's logbook. There were no patient exclusions and, in design, this was a retrospective matched cohort study. Data were retrospectively collected and anonymised from archived patient records, electronic *Epic*™ production software patient records and supporting correspondence. A total of 442 patient cases were studied, including 35 salvage reconstructions (SR) and 35 chronologically consecutive non-salvage reconstructions (NSR). The SR group consisted of patients who underwent free flap breast reconstruction following previous implant-based breast reconstruction. Patients undergoing free flap breast reconstruction for the first time comprised the control group (NSR). These patients had not undergone any type of breast reconstruction previously. Due to the long timeframe of the study, it was important to exclude any learning curve and surgical technology advances, and this also completely avoided selection bias. Therefore, for each SR, the control was selected as the chronologically consecutive NSR, occurring between 2 and 23 days after each SR.

The study design was assessed by the Ethics Committee of the Research & Development Department of Cambridge University Hospitals NHS Foundation Trust and it waived the requirement for Ethics Approval.

### Surgical data

Patient demographics including age, body mass index (BMI), smoking status, and co-morbidities were recorded. In both groups, detailed oncological background was examined, such as whether the patient received post-mastectomy radiotherapy (PMRT).

Details about the patient's initial breast reconstruction were recorded such as age at that surgery, time elapsed and number of revisions prior to SR. Indications for SR were categorized as either objective or subjective. Objective indications included implant rupture, severe CC (Baker III/IV), post-radiotherapy skin and soft tissue changes as well as infection. Subjective indications entailed patient-reported symptoms such as pain, tightness and/or perceived poor cosmesis. The free flap operative details comprised specific flap type, flap weight, operative time, estimated blood loss, intra-operative haemoglobin changes, need for transfusion, flap ischemia time, number of venous anastomoses performed and venous coupler sizes.

Post-operative complications were classed as either major or minor. Major complications denoted operative intervention such as return to theatre for flap exploration, flap loss/necrosis or readmission to hospital. Minor complications were those treated conservatively e.g. seroma aspiration, wound breakdown, oedema and cellulitis, etc.

## Statistical analyses

Statistical analysis was carried out using R3.5.2 software whereby a p-value of <0.05 was deemed to be statistically significant. Data were analysed using unpaired student's t-test and one-way ANOVA for continuous variables or χ2 test for categorical variables.

## Results

### Patient demographics

Between January 2005 and September 2019, a total of 442 patients underwent abdominal free flap breast reconstructions, 35 (7.9%) of these were SR. Of these 35 SR patients six had bilateral (see example in Fig 1) whilst 29 underwent unilateral (see example in Figs 2 and 3) procedures making a total of 41 free flap reconstructions (Table 1). The numbers of patients with the co-morbidities of diabetes, body mass index (BMI) over 30, smoking and hypertension are given in Table 1. Mastectomy pattern and breast cancer type are also described as well as initial breast reconstruction type and any operative history at the donor site (Table 1). The most notable difference between the two groups was in the radiotherapy exposure prior to the

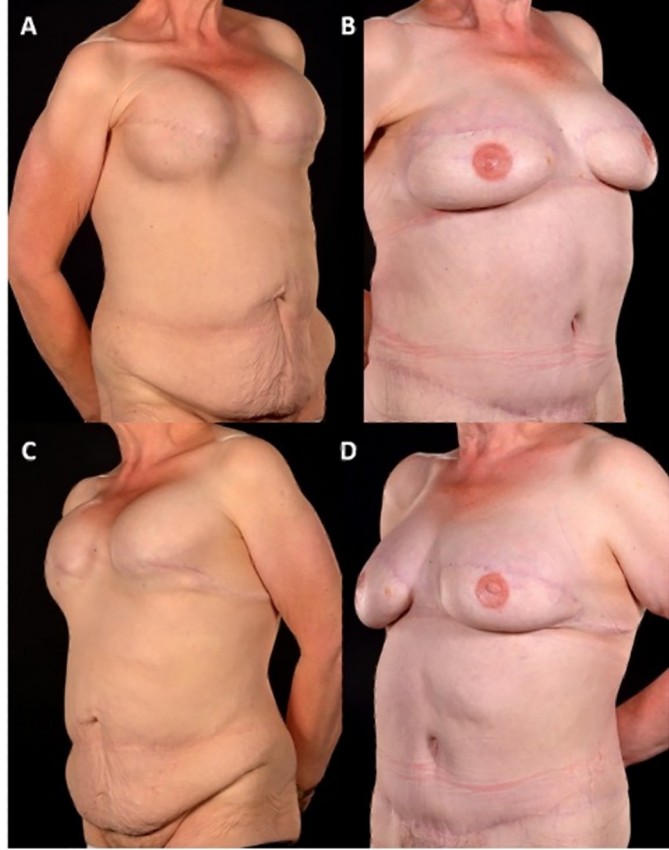

**Fig 1. 55-year-old with bilateral SIEA flap salvage breast reconstruction.** A 55-year-old patient with bilateral SIEA flap salvage breast reconstruction shown before (Fig 1A and 1C) and 2 years post-operatively (Fig 1B and 1D) with natural breast shapes restored, improved positions of the breast mounds and clearly defined inframammary folds. The indications for salvage surgery were grade IV capsular contracture, ruptured silicone implant and poor cosmesis (Fig 1A and 1C).

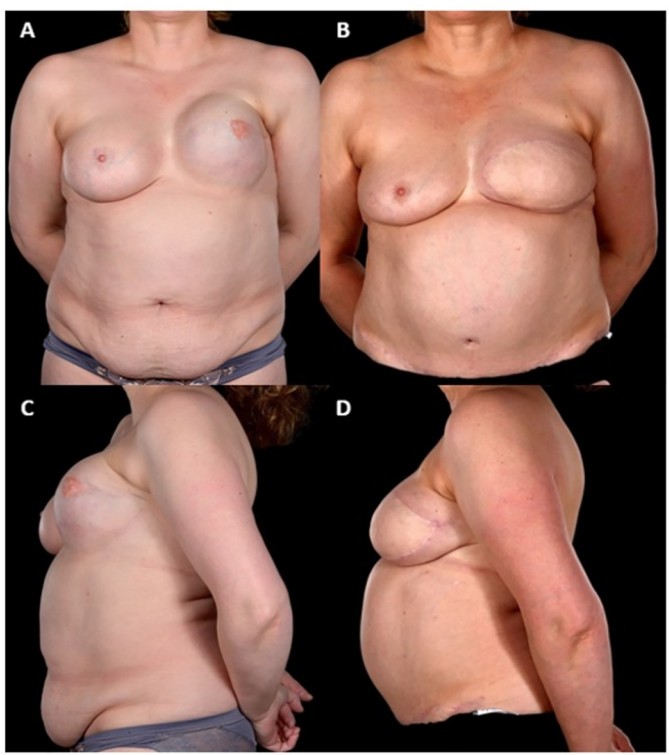

**Fig 2. 46-year-old with left unilateral salvage breast reconstruction with DIEP.** This 46-year-old patient underwent a left unilateral salvage breast reconstruction with a DIEP flap 6 years after an immediate breast reconstruction with implant-only technique. The indications for salvage were grade III capsular contracture with significant pain, poor cosmesis, and a very high-riding implant (status post-radiotherapy) (Fig 2A and 2C). Appearance at 2 years post-operatively showing the reconstructed breast to be lower, wider and more natural (Fig 2B). The inframammary fold has been recreated and is now clearly defined enabled by skin replacement (Fig 2D). The patient was pain-free and very satisfied with the outcome declining further nipple reconstruction.

reconstruction. Notably, almost 90% of (n = 31) of patients in the SR group had received chest wall radiotherapy before their reconstruction, compared to only a quarter (n = 9) in the NSR group (p<0.00001). In the SR group, 42.8% had had immediate breast reconstruction (with implants) at the time of the mastectomy whilst 28.6% had had delayed implant breast reconstruction prior to SR. The comparable figures in the NSR group for the timing of the reconstruction were 68.6% and 20.0% respectively (Table 1). Operative history at the donor site was not significantly different in both groups; 51.4% of patients in both groups had abdominal scars from previous operations.

## Salvage breast reconstruction patient presentation

The mean age of patients at initial breast reconstruction was 45.7±3.0 years and the salvage reconstruction was carried out an average of 7.4±1.9 years later (Table 2). Most patients undergoing SR had had one prior revision of their implant reconstructions. A permanent fixed volume implant had been utilised in 57.1% of patients at the initial reconstruction. Notably, almost 90% of salvage patients had received prior postmastectomy radiotherapy (PMRT) before the SR in contrast to only one quarter of non-salvage patients. Indication was salvage was collected, with most patient sharing two or more indications (Table 2). In 29 patients (83%), the objective indication for the salvage reconstruction was severe CC and most patients

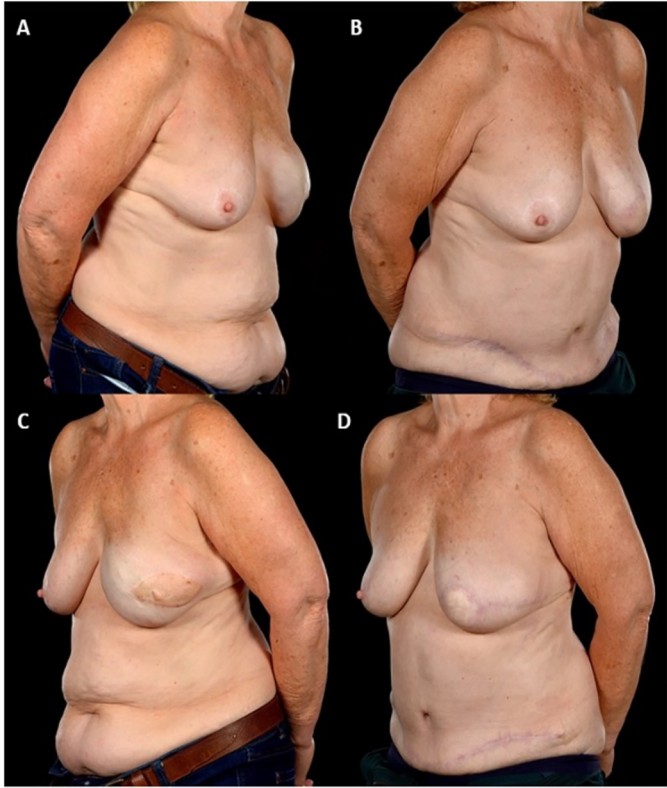

**Fig 3. 73-year-old with left unilateral salvage breast reconstruction with DIEP flap.** This 73-year-old patient underwent a left unilateral DIEP salvage of a previous latissimus dorsi flap-implant reconstruction on account of poor cosmetic results and persistent pain. She is shown before (Fig 3A and 3C) and 9 months following surgery with a better shape of her breast mound and a much better location of her neo-skin paddle (Fig 3B and 3D). The patient declined nipple reconstruction. The indications for salvage were capsular contracture, poor cosmesis (rippling and lateralised skin paddle containing a reconstructed nipple) (Fig 3A and 3C).

cited poor cosmesis and pain as subjective indications for requesting SR (Table 2). 37.1% of patients were documented to have significant post-radiotherapy skin changes, 22.8% of patients had ruptured implants (Table 2) while the implant infection was the cause of the salvage in 8.5%. The operative history at the recipient site prior to SR is shown in Table 2.

## Patient outcomes

The flap types used in both groups were similar with over 70% of all flap types used in each group being the deep inferior epigastric perforator (DIEP) (Table 3). Flap weights (grams ±SEM) were comparable in both groups (Table 3). Unilateral SR took significantly longer to perform that NSR (652±63.45 minutes) and NSR (544±41.48 minutes; p = 0.008) (Fig 4) by almost two hours. This relationship did not hold true for bilateral reconstructions due to the small patient numbers in this subcategory. Contrary to expectation, the average intraoperative blood loss was identical for the two groups. Graphic analysis of estimated blood loss (EBL) (Fig 5) revealed no difference between SR and NSR (p = 0.99) in both unilateral and bilateral reconstructions (Table 3). EBL, however showed a positive Pearson correlation coefficient with duration of surgery (r = 0.4056). Interestingly only one in 20 SR patients (5.7%) required transfusion compared to one in 10 of NSR patients (11.4%). Flap ischaemia times, and vein

**Table 1. Patient demographics for salvage and non-salvage breast reconstructions.**

| Characteristics | Salvage (n = 35) | Non-salvage (n = 35) | p-value |
|---|---|---|---|
| Total flap number | 41 (100%) | 42 (100%) | |
| Unilateral | 29 (70.7%) | 28 (66.6%) | |
| Bilateral | 6 (29.3%) | 7 (33.3%) | |
| Age at operation (years) | 53 ± 3.2 | 50.7 ± 3.1 | 0.32 |
| BMI at operation (kg/m$^2$) | 26.3 ± 0.8 | 26.4 ± 1.1 | 0.93 |
| BMI >30 | 4 (11.4%) | 6 (17.1%) | |
| Co-morbidities | | | |
| **Diabetes** | 2 (5.7%) | 1 (2.9%) | |
| **Smoking status** | | | |
| **Never** | 31 (88.6%) | 28 (80%) | |
| **Current smoker** | 1 (2.9%) | 4 (11.4%) | |
| **Ex-smoker** | 3 (8.6%) | 3 (8.6%) | |
| **Hypertension** | 7 (20.0%) | 5 (14.3%) | |
| **PMRT** | 31 (88.6%) | 9 (25.7%) | |
| **Neoadjuvant** chemotherapy | 20 (57.1%) | 17 (48.5%) | |
| Breast cancer type | | | |
| Ductal carcinoma in situ (DCIS) | 16 (45.7%) | 16 (45.7%) | |
| Invasive ductal carcinoma (IDC) | 9 (25.7%) | 7 (20.0%) | |
| Invasive lobular carcinoma (ILC) | 2 (5.7%) | 6 (17.1%) | |
| Prophylactic | 2 (5.7%) | 3 (8.6%) | |
| Unknown | 6 (17.1%) | 2 (5.7%) | |
| Mastectomy pattern | | | |
| Standard | 11 (31.4%) | 17 (48.6%) | |
| Skin sparing | 9 (25.7%) | 5 (14.3%) | |
| Wise pattern | 7 (20%) | 6 (17.1%) | |
| Unknown | 8 (22.9%) | 7 (20%) | |
| Initial breast reconstruction timing | | | |
| Immediate | 15 (42.8%) | 24 (68.6%) | |
| Delayed | 13 (37.1%) | 7 (20.0%) | |
| Unknown | 7 (20.0%) | 4 (11.4%) | |
| Operative history at donor site | 18 (51.4%) | 18 (51.4%) | |
| Pfannenstiel incision | 9 (25.7%) | 12 (34.3%) | |
| Midline laparotomy incision | 5 (14.3%) | 1 (2.9%) | |
| Open appendicectomy incision | 4 (11.4%) | 2 (5.7%) | |
| No abdominal scars | 17 (48.6%) | 17 (48.6%) | |
| Unknown | 0 | 2 (5.7%) | |

coupler sizes were not significantly different in either group. The additional operative steps unique to SR are summarised in Table 4.

Post-operative complications were classified as either minor or major and shown in Table 3. The difference in overall complication rates between groups was not significant and this was the case for the incidence of complications requiring reoperation (Table 3). The recipient and donor site complications and second touch procedures are further summarised in Table 3. All 83 flap transfers were successful with no total or partial flap losses. A return to theatre was required for two patients following SR and three patients after NSR for flap re-exploration.

**Table 2. Patient presentation in salvage breast reconstruction.**

| Characteristics | Salvage (n = 35) |
|---|---|
| Age at initial breast reconstruction (mean ± SD) | 45.7 ± 3.0 years |
| Time elapsed since initial reconstruction (mean ± SD) | 7.4 ± 1.9 years |
| **Number of surgical reinterventions prior to salvage** | Patients |
| 0 | 4 (11.4%) |
| 1 | 23 (65.7%) |
| 2 | 6 (17.1%) |
| 3 | 3 (8.6%) |
| 4 | 0 |
| 5 | 1 (2.9%) |
| Initial breast reconstruction | |
| Permanent implant | 20 (57.1%) |
| ADM & implant | 7 (20.0%) |
| LD & implant | 5 (14.3%) |
| Expander & implant | 3 (8.5%) |
| Objective indications | |
| Capsular contracture (Baker III/IV) | 29 (82.8%) |
| Post radiotherapy skin changes | 13 (37.1%) |
| Implant rupture | 8 (22.8%) |
| **Implant infection** | 3 (8.5%) |
| Subjective indications | |
| Poor cosmesis (patient & others) | 32 (91.4%) |
| Pain | 30 (85.7%) |
| Operative history at the recipient site | |
| Capsulectomy (often more than once) | 26 (74.2%) |
| Implant exchange | 11 (31.4%) |
| Fat grafting | 5 (14.3%) |
| Other | 2 (5.7%) |

The recipient and donor site complications are summarised in Table 3. Intra-operative anastomotic revision was required in 4 flaps (9.8%) in the SR and 3 flaps (7.1%) in the NSR group (p>0.05). Donor site haematomas were encountered in 2.4% and 7.1% of SR and NSR flaps, respectively. Donor-site wound dehiscence was comparable in the two groups (7.3% SR and 4.8% NSR).

## Discussion

The present study provides the first direct head-to-head comparison of salvage reconstruction (SR) and non-salvage reconstruction (NSR). Previous studies have described series of salvage breast reconstructions with patient numbers ranging from 8 to 191 patients [7, 9, 11, 13, 17]. However, none of these undertook a direct comparison with NSR. Despite the retrospective study design the comparison with NSR cases undertaken over a 15-year period provides an evaluation that can better ascertain the relevant patient risk factors and specific complications attributable to salvage reconstruction. Interestingly, there were few differences in intra-operative and post-operative outcomes between the groups.

Unlike when free flaps are performed during immediate or delayed breast reconstruction, SR have been largely thought of as more technically challenging, taking longer, and beset with greater intra-operative difficulties and post-operative morbidity [13]. The procedure specific

**Table 3. Operative details in salvage and non-salvage breast reconstructions.**

| Parameter | Salvage (n = 35) | Non-salvage (n = 35) | p-value |
|---|---|---|---|
| Free flap type | | | |
| Deep inferior epigastric perforator (DIEP) | 26 (74.2%) | 25 (71.4%) | |
| Superficial inferior epigastric artery (SIEA) | 2 (5.7%) | 3 (8.6%) | |
| MS-TRAM [a] | 4 (11.5%) | 2 (5.7%) | |
| DIEP & SIEA bipedicled | 3 (8.6%) | 3 (8.6%) | |
| Flap weight (g) (n = 29) | 740.50±51 | 812.44±66 | 0.36 |
| Operation time (min) | 659 ±59 | 587 ±49 | 0.07 |
| Unilateral reconstruction (min) | 652 ±63.45 | 544 ±41 | 0.008 |
| Bilateral reconstruction (min) | 778 ±317 | 1,055 ±284 | 0.27 |
| Estimated blood loss (EBL) (ml) | 557 ±132 | 556 ±132 | 0.99 |
| Pearson correlation with operation time | r = 0.43 (p = 0.0108) | r = 0.56 (p = 0.0004) | |
| % Haemoglobin (Hb) change [b] (g/dL) | 18.5 ±2.8 | 18.5 ±3.7 | 0.99 |
| Unilateral % Hb change | 17.3 ±2.9 | 18.3 ±4.3 | 0.069 |
| Bilateral % Hb change | 22.9 ±6.32 | 20.3 ±6.5 | 0.065 |
| Intraoperative transfusion (patients) | 2 (5.7%) | 4 (11.4%) | |
| Flap ischaemia time (min) | 97.6 ±12.4 | 97.5 ±8.14 | 0.49 |
| Unilateral flap | 94.4 ±12.3 | 96.3 ±11.1 | 0.41 |
| Bilateral flap | 115.1 ±25.93 | 97.1 ±11.3 | 0.29 |
| Vein coupler size (mm) (n = 29) | 2.8 ±0.2 | 3.0 ±0.2 | 0.08 |
| **Characteristic** | **Salvage total flap number (n = 41)** | **Non-salvage total flap number (n = 42)** | |
| Complication total number | 20 (48.8%) | 18 (42.9%) | |
| Minor–conservative | 13 (31.7%) | 8 (19%) | |
| Major–surgical | 7 (17%) | 10 (23.8%) | |
| Returns to theatre | 2 (4.9%) | 3 (7.1%) | |
| Flap-related complication | | | |
| Haematoma | 1 (2.4%) | 3 (7.1%) | |
| Anastomotic revision | 4 (9.8%) | 3 (7.1%) | |
| Re-exploration | 2 (4.9%) | 2 (4.8%) | |
| Fat necrosis | 0 | 0 | |
| Partial flap loss | 0 | 0 | |
| Flap loss | 0 | 0 | |
| Donor-site related complication | | | |
| Wound dehiscence | 3 (7.3%) | 2 (4.8%) | |
| Seroma aspiration | 8 (19.5%) | 4 (9.5%) | |
| Dog ear correction | 2 (4.9%) | 2 (4.8%) | |
| Second touch procedures number | 10 (24.4%) | 6 (14.3%) | |
| Seroma aspiration | 8 (19.5%) | 4 (9.5%) | |
| Lipofilling | 2 (4.8%) | 2 (4.8%) | |

[a] MS-TRAM: Muscle Sparing Transverse Rectus Abdominis Myocutaneous flap

[b] Post-operative Hb subtracted by pre-operative Hb divided by pre-operative Hb

operative manoeuvres which contribute to this are summarised in Table 4. Notable amongst these are the capsulectomy, with its attendant bleeding problems, and refashioning of the IMF, both unavoidable parts of improving the cosmetic outcome (a frequent indication for surgery). These two are interrelated since the procedure of excising the tight peri-implant scar tissue further damages (often lowering) the IMF. While certain co-morbidities may be viewed as

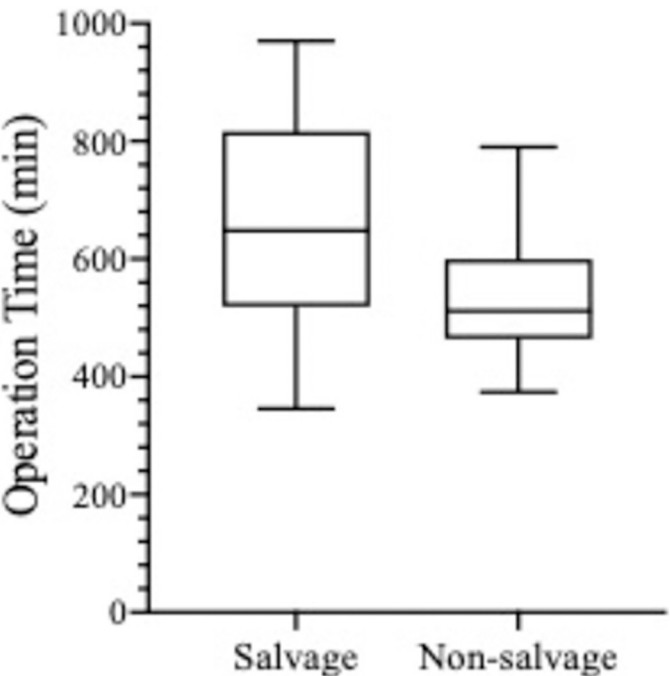

**Fig 4. Operation time for unilateral free flap salvage and non-salvage breast reconstruction.**

contraindications for treatment, our patients were not excluded from operative consideration based on modifiable risk factors. Still, lifestyle advice and smoking cessation guidance was offered pre-operatively to optimise operative fitness. Although it has been postulated that patients undergoing SR are likely to have higher BMIs primarily because of older age at

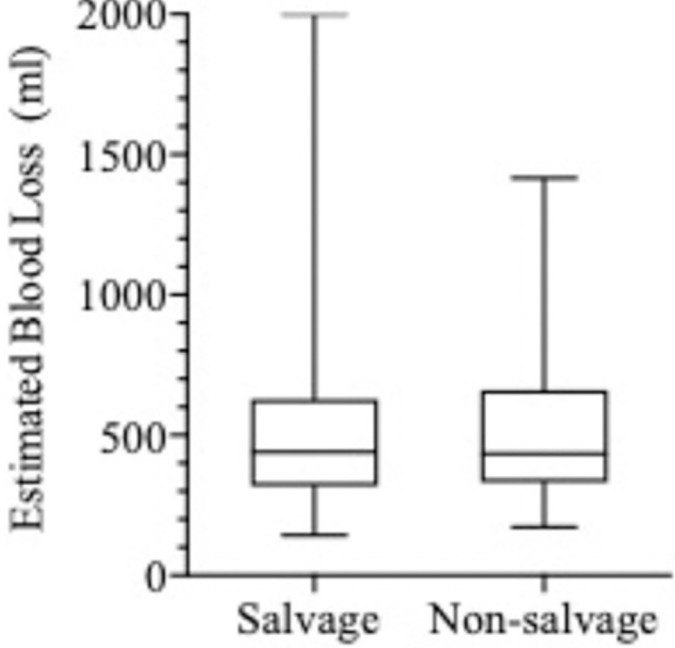

**Fig 5. Estimated blood loss (ml) in salvage and non-salvage breast reconstruction.**

**Table 4. Additional surgical steps undertaken during salvage breast reconstruction.**

| Intraoperative Manoeuvre | Salvage (n = 35) |
|---|---|
| Creation of subcutaneous pocket (from previous subpectoral) | 32 (100%) |
| Capsulectomy | 32 (100%) |
| Excision of irradiated skin and soft tissue | 27 (84.4%) |
| Repositioning of pectoralis major | 26 (81.3%) |
| Internal mammary lymph node excision (to enable IMV exposure) | 25 (78.1%) |
| Refashioning of inframammary fold (recreating the IMF) | 22 (68.8%) |
| Removal of infected breast implant | 8 (25%) |
| Nibbling of rib cartilage to improve access | 8 (25%) |
| Unspecified (no specific detail provided) | 3 (8.5%) |

operation, our study showed no statistical difference in BMI between the groups [13]. Another expectation is that the patients undergoing SR are older as this is often undertaken almost a decade later, our findings support those of others who reported similar results [15].

A notable finding was the significant difference in the prevalence of PMRT; this result parallels that of several studies [7, 21, 22]. Radiotherapy has a two-fold contributory effect to the likelihood of needing SR in the future. Firstly, PMRT after implant-based reconstruction is associated with increased risk of complications in a dose-dependent manner [23]. A meta-analysis of 2348 patients by Ricci et al. suggested that PMRT has greater effect on certain forms of implants over others [24]. Secondly it has also been associated with an increased capsular contracture (CC) risk in all implants [25]. Congruently, in our study, one of the chief reasons for referral for SR following implant-based reconstruction was symptomatic CC. This may explain why, when PMRT is required, autologous reconstruction is considered more favourable than implant-based reconstruction [26].

Other studies have also reported that patients undergoing SR are more likely to have undergone delayed as opposed to immediate reconstruction. Although they have some similarities related to PMRT and its sequelae–namely overall shrinkage of the soft tissue envelope–delayed and SR differ in many respects, as elaborated on in Tables 4 and 5. Subjective indications for SR were reported by the majority of patients, in line with other studies that investigated patient reported outcome measures (PROM) [11, 26]. Although our study did not include assessment of PROMs, evaluating patient satisfaction could be informative. A useful comparison could be with salvage using non-free flap methods such as implant removal combined with serial fat grafting or pedicled latissimus dorsi or perforator flaps.

Consistent with previous studies, our group found the DIEP flap to be the most commonly used in our SR cohort [9, 11, 13]. The DIEP flap has been favoured due to lower donor-site complications, reliability (reduced flap loss) and superior long-term aesthetic outcome as compared with the TRAM flap [26]. Changing the pocket for flap inset entails managing the pectoralis major by suturing it back to the chest wall. This is said to reduce pain associated with the CC as now the muscle has been replaced back to its original position and is no longer under tension. PMRT also causes problems with internal mammary vessel (IMV) exposure, but it remains our preferential recipient site [27]. Whilst this is not unique to SR more of these patients have had PMRT compared to the NSRs. A notable finding was that a significant proportion of SR patients underwent incidental internal mammary lymph node biopsy to expose the recipient IMVs as these nodes were fibrosed from previous surgery and PMRT and become entangled with the surrounding vessels which thus could not be adequately prepared for microanastomoses without removing the lymph nodes [28, 29]. Furthermore, we found a

significant correlation between estimated blood loss (EBL) and operation time (Table 3), indicating that longer operations lead to increased EBL. This reinforces the complexity argument for SR and hence the challenge to obtain good flap outcomes. To achieve this we recommend, amongst other measures, that adequate theatre time is allocated in order to account for these complexities (Tables 4 and 5).

Flap ischaemia is strongly correlated to extent of flap necrosis and post-operative complications, and flap ischaemia time greater than 120 minutes is thought to significantly increase complication risk [30]. There was no significant difference in flap ischaemia and the average time for both groups was under 120minutes. A venous coupler size of less than 2.0mm is known to be related to increased complication risk [31]. Both groups in our study had an average venous coupler size greater than 2.0mm (Table 3). Whilst both groups had a similar flap-related complication rates, the SRs had a proportionately greater rate of donor-site related complications. Although this could be related to flap choice this is unlikely as the was a similar flap distribution in both groups. A possible cause of discrepancy in donor-site morbidity could be due to longer operative time in SR, during which the donor-site may be unattended or open and exposed to the theatre environment for greater durations. Our incidence of major complications was however higher than the 7% reported by Carnevale et al. from La Sapienza in their series of 46 patients in which postmastectomy radiotherapy was found to cause complications perhaps reflecting the complexity of our patients as a tertiary referral centre [32].

## Limitations of our study

Our study cohort size was not sufficient to conduct linear regression analysis, and this may therefore limit transferability of our findings to larger populations. It is also acknowledged that the senior author's experience may attract a specific self-selecting patient cohort seeking SR. Furthermore, the interventions in this study were performed by a single operator which may mean the given results may not be reproducible by other operators. There may also have been an inherent risk of bias in the study design for selection of "controls". However, the sequential consecutive patient selection, whilst not random or matched, aimed to eliminate operator-dependent and team-related improvement of outcomes with experience. Whilst difficult to apply to such research, prospective, blinded randomised studies remain the gold standard. Coriddi et al. collected PROMs for 34 patients out of 137 who underwent SR at six-month follow-up and found statistically significant positive outcomes overall [33]. It may also have been informative to investigate other forms of outcomes such as length of hospital stay, readmission rate and long-term larger population PROMs in future studies. Likewise patient factors (e.g. nulliparity) and hospital stay related factors (e.g. type of intravenous fluid administered) and their role in DIEP flap necrosis would be helpful to investigate in future studies [34]. The retrospective nature of the study also limited in-depth analysis of time dependent factors in DIEP flap reconstruction, the relation to perforator flap anatomy and superficial versus deep venous drainage predominance which have been shown to be important factors in operative time [35].

With the mounting disease burden of breast cancer and commensurate rise in implant-based reconstruction following mastectomy, it is increasingly important to understand the outcomes of SR. Here, we show that SR whilst considered more challenging, can broadly provide similar outcomes to NSR. This is all the more imperative as more operations of this nature will be needed in the years to come. A list of the critical considerations and potential solutions the surgeon should consider is given in Table 5.

Although it is generally held that SR is technically demanding and takes longer, as shown in our study for unilateral cases, its intra- and postoperative outcomes are, however, comparable

**Table 5. Considerations and potential solutions in salvage breast reconstruction.**

| Consideration | Issue(s) | Consequences | Solutions |
|---|---|---|---|
| Previous reconstruction | • Delayed breast reconstruction procedure<br>• Previous lymph node clearance<br>• Subpectoral / Subcutaneous implant<br>• Prior use of Acellular Dermal Matrix (ADM) | • Diffuse tissue scarring / incorporation of ADM<br>• Pectoral fibrosis / contracture<br>• Implant capsular contracture<br>• Lymphoedema<br>• Internal Mammary (IMLN) silicone lymphadenopathy<br>• Risk of cancer recurrence / Breast implant associated lymphoma | • Accurate documentation of surgical history and examination<br>• Pre-operative oncology review<br>• Pre-operative management of lymphoedema<br>Preoperative scans: Computerised tomography (CT)/ Magnetic resonance imaging (MRI) chest and axilla / contralateral mammogram |
| Previous adjuvant therapy | • Irradiated tissue within operative field | • Skin changes: variable including ulceration<br>• Capsular contracture<br>• Other soft tissue fibrosis<br>• Poor tissue planes<br>• Fat necrosis / cysts / seroma<br>• Chest wall osteonecrosis<br>• Friable internal mammary recipient vessels<br>• Delayed healing / wound dehiscence | • Pre-operative CT Angiogram of internal mammary vessels<br>• Meticulous resection of fibrotic tissue<br>Introduce new skin–carefully preserve skin flaps<br>• Total capsulectomy, IMLN biopsy, ± rib sacrifice<br>• Plan for additional autologous volume and skin needs e.g. bipedicled / stacked flap design.<br>• Have "plan B" for microvascular anastomoses / vein grafts / alternate free flap recipient sites<br>• Experienced microsurgeon participation |
| Presenting complaints | • Unfavourable aesthetic appearance<br>• Chronic pain<br>• Multiple failed operations<br>• Previous complications<br>• No inframammary fold (IMF)<br>• Breasts too high | • Skin: contracted / puckered / adherent atrophy<br>• Disrupted anatomical landmarks e.g. IMF<br>• Poor tissue planes<br>• Skin and tissue volume shortage<br>• Fat necrosis / cysts / seroma<br>• Gross breast asymmetry and deformity<br>• Neuromas and neuropathic pain<br>• Increased operative time | • Manage realistic patient expectations<br>• Pre-operative photography: standardized<br>• Pectoralis muscle repositioning<br>• Careful planning of repositioned landmarks<br>• Plan for additional autologous volume needs e.g. bipedicled / stacked arrangement<br>• Consider contralateral balancing procedures<br>• Pain service involvement<br>• Book for adequate theatre time<br>• Recreate IMF with sutures |
| Older age demographic | • ⇩ Cardiorespiratory reserve<br>• ⇧ Thromboembolism risk<br>• Worse healing outcomes<br>• Comorbidities: diabetes, hypertension, cardiac | • Increased anaesthetic risk<br>• Higher risk of post-operative complications e.g. delirium, pulmonary embolism, pneumonia<br>• Increased hospital stay duration | • Medical optimisation and preassessment<br>• Consider high dependency post-operative care<br>• Plan for increased nursing and physiotherapy demands<br>• Strict diabetic / thromboprophylaxis control |

to NSR. SR and NSR were all successful with a similar complication profile. Therefore, SR with free flaps provides a reliable salvage option following failed implant-based reconstructions and should be actively considered when faced with this patient group.

## Supporting information

**S1 Dataset.**
(XLSX)

## Author Contributions

**Conceptualization:** Charles M. Malata.

**Data curation:** Christine Bojanic, Laura J. Fopp.

**Formal analysis:** Christine Bojanic.

**Investigation:** Christine Bojanic.

**Methodology:** Christine Bojanic.

**Project administration:** Bruno Di Pace, Dina T. Ghorra.

**Supervision:** Charles M. Malata.

**Validation:** Charles M. Malata.

**Writing – original draft:** Christine Bojanic.

**Writing – review & editing:** Bruno Di Pace, Dina T. Ghorra, Laura J. Fopp, Nicholas G. Rabey, Charles M. Malata.

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
