## [Decision Letter · Decision Letter 0]

14 Feb 2023

PONE-D-23-00098A Comparison of Presentations and Outcomes of Salvage (Tertiary) versus Non-Salvage (Immediate and Delayed) Free Flap Breast Reconstructions – Results of a 15-year Tertiary Referral Centre ReviewPLOS ONE

Dear Dr. Bojanic,

Thank you for submitting your manuscript to PLOS ONE. After careful consideration, we feel that it has merit but does not fully meet PLOS ONE’s publication criteria as it currently stands. Therefore, we invite you to submit a revised version of the manuscript that addresses the points raised during the review process.

We look forward to receiving your revised manuscript.

Kind regards,

Fabio Santanelli, di Pompeo d'Illasi, MD, PhD

Academic Editor

PLOS ONE

Journal Requirements:

2. You indicated that ethical approval was not necessary for your study. We understand that the framework for ethical oversight requirements for studies of this type may differ depending on the setting and we would appreciate some further clarification regarding your research. Could you please provide further details on why your study is exempt from the need for approval and confirmation from your institutional review board or research ethics committee (e.g., in the form of a letter or email correspondence) that ethics review was not necessary for this study? 

Please include a copy of the correspondence as an ""Other"" file.

Reviewers' comments:

Reviewer's Responses to Questions

**Comments to the Author**

1. Is the manuscript technically sound, and do the data support the conclusions?

Reviewer #1: Yes

2. Has the statistical analysis been performed appropriately and rigorously? 

Reviewer #1: Yes

3. Have the authors made all data underlying the findings in their manuscript fully available?

Reviewer #1: Yes

4. Is the manuscript presented in an intelligible fashion and written in standard English?

Reviewer #1: Yes

5. Review Comments to the Author

Reviewer #1: I would like to commend the authors for their interesting manuscript which provides insight on a peculiar side of breast reconstruction, by sharing their experience as a tertiary referral center. However, the manuscript presents aspects which warrant some revisions. They have been subdivided into sections according to the areas of concern.

Abstract:

- Abstracts should preferably not include abbreviations or acronyms and have a word limit of 300 words for this journal. Your current abstract counts 255 words. Please consider replacing the acronyms with the fully expanded terms.

- The study compared 41 salvage flaps to 42 immediate or delayed flaps for breast reconstruction. Please specify in the “Methods” section of your abstract that all featured patients received an abdominally-based free flap.

- What types of outcomes were assessed? From your description it would seem that you assessed clinical outcomes with operative times, intraoperative blood loss and complication rates. The aim of your study should be stated more clearly in the introductory section of the abstract where you state the aim of your study. Additionally, state the nature of the study you conducted more explicitly.

Introduction:

- I acknowledge your definition of “tertiary” or “salvage breast reconstruction” which of course can be traced back to Hamdi et al.’s paper from 2010 (ref. 13). This concern is purely on a semantic standpoint: some consider “secondary breast reconstruction” as the “reconstructive procedure to correct complications and to improve the aesthetics when a patient is dissatisfied with her initial reconstruction” (Hee Chang Ahn et al. in J Korean Soc Plast Reconstr Surg. 2009), as opposed to primary reconstructions. Others interchangeably use “primary” as “immediate”, and “secondary” as “delayed” procedures, as did Hamdi et al. in their 2010 paper. My best recommendation is to either clear the definitions you chose in the introduction of your manuscript, or to scrape “primary”, “secondary” and “tertiary” altogether (including from the title), referencing the confusion in definitions, and just use “salvage” and “non-salvage” (i.e. immediate and delayed).

- The introduction is otherwise well-thought and eases into the subject appropriately. However, consider implementing the following information to strengthen your concepts: o Breast implant lifespan has been estimated around 10 years, with patients potentially undergoing revisional surgeries as often as 4 times or more in their lifetime (PMID: 36376583).

o Regarding silicone extravasation, recent evidence found breast implant ruptures to be extracapsular in many as 25.8% of cases (PMID: 36229658).

Material and Methods:

- Please clarify whether the study you conducted is “simply” a retrospective chart review or specifically a case-control study. Regardless of which, ensure that your paper is prepared in accordance with STROBE guidelines.

- It appears that the patients from the non-salvage group were chosen chronologically. Were they still matched to the salvage group according to patient demographics (i.e. age and BMI)? It would seem that way as suggested by the p-values in table 1 showing that the differences between groups were not statistically relevant. This has not been explicitly stated in your manuscript, and should be addressed.

- The justification of why you only included procedures performed by the senior author to exclude the effects of learning curve (“Due to the long timeframe of the study […] selection bias.” In Materials and methods, “Patient sample” subheading, p. 4, lines 100-102) has no place in this portion of the manuscript and should be moved in the discussion. Additionally, your study features bilateral cases as well. Was a 2-team approach used for those cases or were they also performed by a single surgeon? If a second surgeon was used, would you say that the difference in experience between the senior surgeon and a hypothetical second surgeon could have affected the results in this group due to a difference in learning

curve?

- Additionally, inclusion and exclusion criteria should be listed more clearly and explicitly.

- Consider changing your terminology in the manuscript as well as in Table 3, by replacing “partial flap necrosis” with “fat necrosis” according to the Rao grading system for consistency (PMID: 25415090).

Discussion:

- Your salvage and non-salvage groups were matched according to age and BMI alone. Why could they not be matched according to their comorbidities, mastectomy type and donor site characteristics as well? Please address this as a potential limitation.

- You reported no case of either partial flap loss or fat necrosis in your study. While this is impressive, consider addressing some of the identified predictive and protective factors for partial necrosis in DIEP flap breast reconstruction (PMID: 23851375).

- Most patients received a DIEP flap. It would be interesting for the authors to briefly share their reconstructive ladder in terms of salvage options according to patient characteristics (i.e. breast size volume, donor site characteristics and laterality).

- Post-mastectomy radiotherapy has indeed been found to cause complications following beast reconstruction, with a study from Carnevale et al. reporting major complications and the need to reoperate in 7% of their 46-patient cohort (PMID: 23801395). Would you say that this complication rate is higher in your institution? Could that be biased by the fact

that yours is a tertiary referral center? Please address this in your discussion.

- When assessing clinical outcomes of your abdominally-based free flaps, were time- dependent factors in DIEP flap breast reconstruction taken into consideration? Is has been demonstrated that increased flap weight, related perforators number, and venous drainage negatively influence operative time (PMID: 28758229).

- The authors discuss using the internal mammary vessels as recipient vessels, which at times can be damaged from radiotherapy. Could having used recipient vessels in the axilla been a viable alternative if not even the recipient area of choice especially in these cases? (PMID: 24782202).

- This study did not account for type and volume of breast implants originally placed in patients from the salvage group. Consider addressing this as a possible shortcoming to your study and reference this study from Zhao et al. which found larger-sized implants to be more commonly associated with salvage reconstruction and conversion to abdominally- based free flaps (PMID: 29734445).

6. PLOS authors have the option to publish the peer review history of their article (what does this mean?). If published, this will include your full peer review and any attached files.

Reviewer #1: **Yes: **Guido Firmani

---

## [Author Response · Author response to Decision Letter 0]

31 May 2023

Please kindly see our reply to the reviewers in separate Word document, included in the submission and titled "cover letter - responce to reviewers". Many thanks.

---

## [Decision Letter · Decision Letter 1]

26 Jun 2023

A comparison of presentations and outcomes of salvage versus non-salvage abdominal free flap breast reconstructions – results of a 15-year tertiary referral centre review

PONE-D-23-00098R1

Dear Dr. Bojanic,

We’re pleased to inform you that your manuscript has been judged scientifically suitable for publication and will be formally accepted for publication once it meets all outstanding technical requirements.

Kind regards,

Fabio Santanelli, di Pompeo d'Illasi, MD, PhD

Academic Editor

PLOS ONE

Additional Editor Comments (optional):

Reviewers' comments:

Reviewer's Responses to Questions

**Comments to the Author**

1. If the authors have adequately addressed your comments raised in a previous round of review and you feel that this manuscript is now acceptable for publication, you may indicate that here to bypass the “Comments to the Author” section, enter your conflict of interest statement in the “Confidential to Editor” section, and submit your "Accept" recommendation.

Reviewer #1: All comments have been addressed

2. Is the manuscript technically sound, and do the data support the conclusions?

Reviewer #1: Yes

3. Has the statistical analysis been performed appropriately and rigorously? 

Reviewer #1: Yes

4. Have the authors made all data underlying the findings in their manuscript fully available?

Reviewer #1: Yes

5. Is the manuscript presented in an intelligible fashion and written in standard English?

Reviewer #1: Yes

6. Review Comments to the Author

Reviewer #1: The authors addressed all the areas of concern successfully, and I am satisfied with their response. I congratulate the authors for their interesting research.

7. PLOS authors have the option to publish the peer review history of their article (what does this mean?). If published, this will include your full peer review and any attached files.

Reviewer #1: **Yes: **Guido Firmani

---

## [Editor Report · Acceptance letter]

14 Jul 2023

PONE-D-23-00098R1 

A comparison of presentations and outcomes of salvage versus non-salvage abdominal free flap breast reconstructions – results of a 15-year tertiary referral centre review 

Dear Dr. Bojanic:

I'm pleased to inform you that your manuscript has been deemed suitable for publication in PLOS ONE. Congratulations! Your manuscript is now with our production department. 

Kind regards, 

on behalf of

Prof. Fabio Santanelli, di Pompeo d'Illasi 

Academic Editor

PLOS ONE